# The Impact of Social Media on Children’s Mental Health: A Systematic Scoping Review

**DOI:** 10.3390/healthcare12232391

**Published:** 2024-11-28

**Authors:** Ting Liu, Yanying Cheng, Yiming Luo, Zhuo Wang, Patrick Cheong-Iao Pang, Yuanze Xia, Ying Lau

**Affiliations:** 1Faculty of Applied Sciences, Macao Polytechnic University, Macao 999078, China; p2314982@mpu.edu.mo (T.L.); p2417889@mpu.edu.mo (Y.L.); p2418258@mpu.edu.mo (Y.X.); 2School of Business and Management, Jilin University, Changchun 130015, China; chengyanring@foxmail.com; 3School of General Education, Beijing Normal University-Hong Kong Baptist University United International College, Zhuhai 519087, China; zhuowang@uic.edu.cn; 4The Nethersole School of Nursing, The Chinese University of Hong Kong, Hong Kong 999077, China; yinglau@cuhk.edu.hk

**Keywords:** social media, children, mental health, systematic scoping review

## Abstract

Background: In the digital age, safeguarding children’s mental health (CMH) has emerged as one of the most pressing challenges. The rapid evolution of social media (SM) from a basic networking platform to a multifaceted tool has introduced numerous conveniences. However, it has also posed significant challenges to children’s mental well-being. Methods: Given the intricate relationship between the widespread use of SM and mental health issues in children, this study conducted a systematic scoping review to examine the literature on the impact of SM on CMH from 2014 to 2024. Literature searches were performed across five databases (Web of Science, ScienceDirect, Scopus, PubMed, and APA PsycInfo), and the retrieved studies were screened, extracted, and analyzed by the Preferred Reporting Items for Systematic Reviews and Meta-Analysis Extension for Scoping Review (PRISMA-ScR) guidelines. Results: The review identified a complex relationship between SM use and CMH. Different SM platforms exhibited varying effects on children. Frequent SM use was strongly associated with lower self-esteem, depressive symptoms, anxiety, and other mental health challenges in children. Conversely, moderate use of SM facilitated social interactions and emotional expression, which may have a positive influence on mental health. Additionally, factors such as social support from family and school were found to play a critical role in mitigating the negative effects of SM on CMH. Conclusions: To enhance CMH, it is essential to guide children in the appropriate use of SM, promote awareness of privacy protection, and ensure adequate family and social support. Future research should further investigate the specific mechanisms underlying SM use and its differential effects on children across varying age groups and regions.

## 1. Introduction

Protecting CMH is one of the most pressing challenges in the current digital age. SM have rapidly evolved from a simple social networking platform to a multifunctional tool utilized across various domains, with over 5 billion global users as of 2023, representing more than 63% of the world’s population [1]. Despite the numerous conveniences associated with this growth, CMH faces significant challenges in an increasingly digitalized environment [2]. Research indicates that the prevalence of mental disorders among children continues to rise. These issues are often not diagnosed or treated promptly [3]. The personalized, interactive experiences and constant accessibility offered by SM are particularly appealing to children whose self-concept is still developing [4,5]. The immersive design of these platforms enhances their attractiveness, making them more engaging and potentially addictive for young users. Frequent use of SM has been strongly linked to the development of psychiatric disorders in children. This association can be attributed to their still-maturing self-regulatory abilities, which may lead to increased anxiety and agitation [6,7].

“Social media”, as an electronic communication tool that enables users to create, share, and interact with each other through internet platforms, began to take shape in the 1990s and rapidly gained popularity in the 2000s with the emergence of social networks such as Facebook and Twitter [8]. Contemporary SM encompasse a variety of forms, including social networks, content-sharing communities, and virtual worlds, offering a rich and diverse array of online communication methods [9,10]. Recent statistics indicate that approximately 70% of children aged 8–12 use SM, despite age restrictions on many platforms, highlighting the pervasive influence of these tools [11]. Moreover, on average, children aged 9–12 spend 1.5 h per day on social platforms, with YouTube and TikTok being the most popular among this demographic [12].

This evolving form of communication provides children with abundant opportunities for socialization and has prompted extensive research into its impact on CMH. Studies have revealed that excessive SM use is associated with negative mental health outcomes in children, including heightened levels of anxiety, depression, and poor sleep quality [13,14]. A meta-analysis found that children who spend more than two hours per day on SM are at a significantly higher risk of experiencing symptoms of anxiety and depression compared to their peers with limited screen time [15].

CMH issues frequently exhibit hidden and complex characteristics, making them challenging for parents and educators to detect. For instance, depressive symptoms may manifest as a persistent low mood, diminished interest in daily activities, irritability, and mood swings, which can stem from the adaptive challenges associated with the school environment as well as familial stressors [16]. A study revealed that children aged 10–12 who experienced cyberbullying reported higher levels of social anxiety and depressive symptoms, underscoring the psychological toll of online harassment [17].

Symptoms of anxiety are also prevalent among children, particularly in the context of SM exposure. A longitudinal study demonstrated that children who frequently engage with SM before bedtime are more likely to experience sleep disturbances, which in turn contribute to elevated anxiety levels [18]. Symptoms often include excessive worry, difficulty concentrating, and disrupted sleep patterns, which are commonly misattributed to behavioral issues rather than underlying anxiety [19].

SM addiction has become a growing problem, especially among children. Frequent or excessive SM use usually refers to users spending a lot of time online or using social platforms at a very high frequency. However, different studies have used a variety of measures (e.g., parent reports, questionnaires, and app monitoring), leading to differences in the definition of SM frequency [20]. Studies have defined SM use of more than 3 h per day as “excessive” use [21], while others considered checking SM more than 10 times per day or for more than 2 h as “frequent” use [22]. In the case of children, the definition of “frequent” SM use is typically considered as more than 1 h per day [23], while “moderate” use is often defined as up to 30 min per day, provided it does not interfere with sleep, learning, or socialization [24]. These differing definitions contribute to the complexity of assessing the impact of SM use on CMH.

Against this backdrop, an increasing body of research has sought to investigate the relationship between SM use and CMH. On one hand, the frequent use of SM may contribute to heightened anxiety and depressive symptoms in children [22]. SM can also have a positive impact on CMH by providing emotional support and enhancing social connections [25]. Specifically, the positive effects of SM are most evident in children’s active engagement behaviors, such as posting content, participating in discussions, and interacting with peers, which can foster social involvement and emotional expression. In contrast, excessive passive use (e.g., merely viewing others’ content) is likely to provoke social comparisons, making children more susceptible to low self-esteem and anxiety [26]. SM use may also influence children’s self-esteem and body image, potentially leading to negative emotional outcomes [27].

Meanwhile, some studies grounded in ecosystem theory suggest that SM interactions with children’s immediate environments (e.g., home and school) may result in complex mental health outcomes [28]. These varied findings not only reflect the heterogeneous and unsystematic nature of the current research but also highlight methodological shortcomings and issues related to construct validity [29]. The causal effects of SM on CMH and its actual impact cannot be conclusively inferred at this time. There is an urgent need for more systematic and in-depth research to clarify the roles and mechanisms of SM in CMH.

This study reviewed the existing literature related to the impact of SM on CMH. Of the seven reviews analyzed, four were systematic reviews, two were scoping reviews, and one was a literature review. These reviews investigated the relationship between SM use and various mental health issues, including depression, anxiety, addictive behaviors, and cyberbullying. Previous reviews have established that the excessive use of SM is strongly associated with mental health problems and that prolonged exposure may predispose individuals to depression and anxiety, while moderate use may yield positive effects. However, several limitations in the previous reviews were identified (see Appendix A). First, some reviews utilized fewer than five databases in their literature searches, which limits the comprehensiveness and reliability of their findings [30,31,32,33]. Second, the literature included in these reviews was often not sufficiently current to capture new developments and trends [31,34,35], and these reviews frequently did not specify the range of years covered by their literature searches, potentially affecting the timeliness and relevance of their findings [36,37]. Additionally, many of these reviews focused predominantly on articles from the short-term period of the COVID-19 pandemic, which may not accurately reflect long-term trends [30]. Some reviews were limited in scope, concentrating solely on negative effects or restricting their analysis to depression, thereby failing to adequately evaluate the multidimensional impact of SM on CMH [31]. Finally, a number of reviews did not adequately differentiate the effects on various age groups, often analyzing data from children, adolescents, and adults in a mixed manner, which may obscure characteristics specific to each age group [33,36].

The purpose of this study is to comprehensively summarize the current state of research focusing on the age-specific impacts of SM on CMH by systematically reviewing the relevant literature from 2014 to 2024. This review aims to elucidate the multifaceted effects of SM use on CMH, integrating the existing research while sorting through its complexity, diversity, and contradictions. This research will attempt to address the following questions:(1)What countries or regions are the primary focuses of research on the impact of SM on CMH?(2)What are the main research themes regarding the effects of SM on CMH?(3)Which SM platforms have the most significant influence on CMH?(4)What factors govern the impact of SM on CMH?(5)What unique challenges, impacts, and benefits do children encounter in their use of SM?

This study intends to fill the gaps identified in the existing reviews concerning the impact of SM on CMH and to provide an authoritative and comprehensive analysis of how SM affects the mental health of children aged 6 to 13 years. By conducting a systematic review, the research aims to assist policymakers, educators, and parents in better identifying and addressing these impacts, ultimately improving strategies for managing CMH.

## 2. Methods

The methodology adopted in this study was a systematic scoping review designed to identify research findings concerning the association between SM use and mental health status among children aged 6 to 13 years. This review adhered to the guidelines set forth by the PRISMA-ScR guidelines [38], aiming to enhance the methodological quality and credibility of the empirical data obtained and generated [39].

### 2.1. Search Strategy

This study screened relevant literature from five databases: Web of Science, ScienceDirect, Scopus, PubMed, and PsycINFO. The search period spanned from 2014 to 2024, with the search conducted on 16 August 2024. The search terms included “social media”, “children”, and “mental health”, among others. The search strategy involved the following steps: (1) an initial screening of all databases to exclude irrelevant studies; (2) de-duplication of records using EndNote 21.4 software; (3) filtering based on titles and abstracts to further narrow down the literature; and (4) a final review of the full text to exclude studies not directly related to the research topic. The specific search formula is presented in Table 1.

### 2.2. Data Selection and Extraction

The inclusion and exclusion criteria for this review were structured using the PICO framework to ensure a systematic and focused selection of studies. These criteria are summarized in Table 2, highlighting key elements such as the target population, intervention focus, and expected outcomes.

### 2.3. Data Charting

A data extraction form was developed based on the methodological guidelines for scoping reviews provided by the PRISMA-ScR guidelines [38]. The form was further adapted after piloting with five articles. It includes the following items: author, year, country, study type, topic, mental health indicators, sample size, age, sample country/region, SM platform, influencing factors/impacts, and main findings. Data were extracted by two independent reviewers, and any disagreements were resolved through consultation with a third reviewer.

### 2.4. Collating, Summarizing, and Reporting the Results

Data were collected, summarized, and analyzed using descriptive statistics to characterize the sample articles. The descriptive findings were presented through graphs and charts. The results were interpreted through a narrative synthesis that addresses the research questions posed in the review, and this interpretation was validated by all authors.

## 3. Results

As shown in Figure 1, the systematic search yielded a total of 6714 articles. Of these, 871 duplicate records were identified and deleted using EndNote. Additionally, 11 records were flagged as ineligible by an automated tool due to publication date criteria, and three records that did not meet language requirements were also removed.

After these records were excluded, a total of 5829 records were screened in accordance with the PRISMA-ScR guidelines (see Appendix B). The review of article titles and abstracts led to the exclusion of 5746 articles that were not directly related to the research topic. Of the excluded articles, approximately 71% pertained to unrelated topics, such as child sexual abuse, school shootings, the three-child rearing policy, the role of SM in pediatric surgery, social networking as a tool for authentic social interaction, and the use of SM during the COVID-19 pandemic. Approximately 20% of the excluded records addressed the wrong population, primarily consisting of parents, adolescents older than 13 years, and children with medical conditions. Additionally, around 9% of the excluded studies utilized SM tools for participant recruitment or for administering questionnaires.

In addition to five irretrievable reports, the remaining 76 articles were evaluated. Among these, 54 were deemed irrelevant, covering topics such as children’s data privacy, the relationship between SM and climate change, ethical analyses of SM terminology, the impact of SM challenges on children’s toxic intake behaviors, the influence of parents’ “sharing parenting experiences” on children’s use of SM, and the impact of face-to-face relationships when children use SM. Three articles were excluded because they focused on adolescents over the age of 13. Finally, one article addressed the wrong population of “social media users”, while two articles were excluded for exclusively focusing on dangerous games related to SM and screen media use, respectively. Ultimately, a total of 16 articles were included in the assessment of this systematic review (see Table 3 [37,40,41,42,43,44,45,46,47,48,49,50,51,52,53,54]).

### 3.1. Geographical and Sample Characteristics

Of the 16 studies reviewed, the United States, Canada, and Australia had the highest number of publications. As illustrated in Figure 2, the United States led with five studies, indicating its significant contribution to research on the impact of SM on CMH. Regarding geographic distribution, six studies were conducted in Europe and four in North America, both of which featured larger sample sizes. The usage rates of SM in the United States and Canada are notably high on a global scale, which may have led researchers to focus more on the behaviors and mental health issues of users in these regions, particularly in relation to children [55]. Furthermore, the relatively less stringent regulation of SM in the United States compared to other countries could also contribute to an increased emphasis on research regarding how these platforms affect CMH [56]. This suggests that these regions have placed a greater emphasis on studying the impact of SM on CMH. In contrast, relatively few studies were conducted in the Asia–Pacific region, highlighting a potential opportunity for further research. Additionally, three studies did not specify a sample region. The geographical distribution and sample details are presented in Table 4.

### 3.2. Year of Publications

As illustrated in Figure 3, the year 2020 marked the highest number of publications, with five studies. This increase may be linked to heightened interest in SM during the pandemic. However, some studies published in 2020 may have used data collected pre-pandemic, so this trend should be interpreted with caution. The rise in publications in 2024 indicates that researchers continue to focus on the impact of SM on CMH, particularly regarding emerging platforms. In contrast, the relatively fewer publications from 2011 to 2019 suggest a lack of early attention to this important issue. In conclusion, the impact of SM on CMH has gradually emerged as a significant area of academic research since 2020.

### 3.3. Types of Studies

Among the 16 studies, cross-sectional studies were the most commonly employed methodology, comprising a total of eight studies. This was followed by empirical studies and longitudinal studies, each represented by two studies. Additionally, there were two clinical reports, while both technical reports and exploratory studies were less common, with one of each category, as illustrated in Figure 4.

### 3.4. Research Topics

Among the 16 studies, the research topics can be categorized into five themes: (1) mental health issues, which include depression and subjective well-being; (2) digital literacy and risk awareness, covering aspects of digital literacy, SM risks, and threats; (3) cyber security and cyberbullying, focusing on cyberbullying and preventive guidance related to SM; (4) social and emotional development, encompassing social and emotional development, interpersonal relationships, and social skills; and (5) family and society, which includes parental supervision, social determinants, and gender equality. Details of these themes are presented in Table 5.

### 3.5. Indicators of the Impact of SM on CMH

The impact of SM on CMH is multifaceted, encompassing emotional states, social factors, quality of life, body-related indicators, and physiological factors. The association with emotional states was the most prominent, with nine occurrences, including depressive symptoms, social anxiety, and self-esteem levels. This underscores the significance of emotional states in this context. Social factors and physiological factors each appeared six times, highlighting the importance of social support systems and the interplay between physical and mental health, respectively. In contrast, body-related indicators and quality of life associations appeared less frequently, with five and four occurrences, respectively, indicating that while these areas are important, they have received somewhat less attention in the current study. Details of these categories are presented in Table 6 and Figure 5.

### 3.6. SM Platforms

Across the 16 pieces of literature, mentions of SM platforms predominantly focused on social networking platforms (e.g., Facebook, Instagram), which were referenced 12 times, and video-sharing platforms (e.g., YouTube), mentioned 8 times. This highlights the significant impact of these platforms on CMH. Instant messaging applications (e.g., Snapchat, WhatsApp) were cited five times, reflecting their role in facilitating social interactions. Details of these categories are presented in Table 7 and Figure 6.

### 3.7. Factors and Influences

An analysis of the 16 studies revealed that the impact of SM on CMH is governed by multiple factors, with significant individual differences, as illustrated in Table 8 and Figure 7. First, gender (32.14%) emerged as the most dominant influencing factor, suggesting that boys and girls may experience and respond to SM usage differently, necessitating individualized support measures. Secondly, age (28.57%) also plays a crucial role in shaping children’s responses to SM, highlighting the distinct challenges that children of various ages encounter in understanding and utilizing these platforms. Social support (17.86%), as the third most significant factor, underscores the importance of family, school, and peers in either mitigating or exacerbating the psychological impact of SM on children. Additionally, parental influence (14.29%) emphasizes the essential role of parents in guiding and regulating their children’s use of SM through understanding, monitoring, and controlling their engagement with these platforms. While economic status, cultural factors, and psychological and emotional aspects represent relatively minor influences (7.14% each), they still contribute to the complex ecosystem that shapes the impact of SM on CMH.

## 4. Discussion

### 4.1. Main Findings and Results of Studies

The research indicates that SM use has complex and multifaceted effects on CMH, moderated by a variety of factors, including gender, age, and social support. Specifically, the use of platforms such as YouTube and Instagram is significantly associated with decreased body satisfaction and an increased eating pathology among children, particularly those who frequently engage in appearance comparisons. However, not all SM use yields negative outcomes. When children have access to parental supervision, family support, and the capacity to cultivate self-compassion while engaging with SM, these factors serve as protective barriers that effectively mitigate the adverse effects of problematic SM use and cyberbullying. While SM can facilitate social connections among children to some extent, excessive use is strongly linked to decreased well-being, increased risky behaviors, and heightened emotional challenges. This underscores the importance of maintaining a balance between SM use and family engagement in online activities.

### 4.2. SM and Family, Social Support

Family and social support play a crucial moderating role in children’s SM use. According to ecosystem theory, children are situated within multiple nested environmental systems that interact and collectively influence their growth and development [57]. Parental involvement and supervision, as key factors within the micro-system, can effectively mitigate the potential negative impacts of SM by establishing rules and jointly participating in SM activities [54].

Simultaneously, social support functions within immediate environmental systems, such as family and school, to alleviate the negative effects of cyberbullying and problematic SM use. Frequent SM engagement is associated with positive friendships, particularly significant among girls; however, problematic use is often correlated with poorer family relationships and diminished social support, leading to social isolation and increased emotional stress [42]. Ecosystem theory further emphasizes that SM, as an emerging external system, is intricately connected to the family and school environments where children reside. It influences CMH and social behaviors through unique communication styles. For instance, negative experiences on SM, such as cyberbullying, may have their impact on children’s well-being mitigated or exacerbated by family and school support systems [42].

Moreover, the quality of social support is particularly critical within this complex network of environments. High levels of social support not only buffer the risks posed by SM but also promote CMH [43]. On this basis, comprehensive mental health programs at the school level are vital for mitigating the negative impacts of SM on CMH. Implementing systematic mental health monitoring and intervention strategies in educational settings is recommended [57].

Future research should concentrate on developing practical parent training programs to equip parents with effective SM supervision skills, such as limiting screen time and participating in online activities together, to reduce the psychosocial impact of online risks on children. Additionally, schools can introduce a systematic SM education curriculum that teaches students to identify and respond to cyberbullying and maintain healthy online social behaviors, and that provides teachers with resources to identify and intervene in cyber risks. These intervention strategies can contribute to co-constructing support systems at both the family and school levels that promote CMH development.

### 4.3. SM Use and Subjective Well-Being

Subjective well-being (SWB) refers to an individual’s overall assessment of his or her quality of life and encompasses multiple dimensions, including positive emotions, negative emotions, and life satisfaction. The impact of SM on children’s SWB is multidimensional and bidirectional. On the one hand, SM provide a platform for children to stay connected with their friends, which can alleviate feelings of loneliness and anxiety to some extent, thereby enhancing their positive emotions and life satisfaction. On the other hand, research indicates that over-reliance on SM can exacerbate children’s negative emotions, leading to increased depressive symptoms, anxiety, and psychological stress, ultimately resulting in a significant reduction in their SWB [46].

Additionally, studies have found that the relationship between SM use and SWB is influenced by various factors, including gender, family support, and family income. For instance, girls tend to experience a greater decline in SWB compared to boys, while strong family support and higher family income are associated with better SWB outcomes [54]. To effectively enhance children’s SWB, it is recommended that parents and educators actively guide children in the mindful use of SM. This includes encouraging meaningful social interactions and strengthening family support systems, particularly for girls, to help mitigate negative influences such as cyberbullying and social comparisons.

### 4.4. Roles of Parents and Educators

The role of parents in guiding children’s SM use is crucial. Numerous studies indicate that parental accompaniment, communication, and supervision during SM activities can significantly mitigate the negative impacts of unhealthy usage behaviors on CMH. For example, a parental presence and open communication can effectively prevent issues such as cyberbullying, while promoting positive social interactions among children [43,50]. Parents should strive to be media-literate, enabling them to help their children recognize the risks and opportunities associated with SM. Parental education and emotional support can significantly enhance a child’s emotional resilience in the face of online stress and social comparisons [54].

Schools also play an equally vital role in fostering healthy SM habits among children. By implementing media literacy and mental health programs, teachers can equip children with skills to cope with SM-related stress and encourage positive online interactions [41]. Additionally, schools should provide a supportive environment that establishes healthy digital norms and alleviates feelings of isolation and anxiety through relationship guidance [48]. Interpersonal support within schools, particularly positive interactions between teachers and students, can help compensate for the emotional support that some students may lack at home [45].

Collaboration between parents and educators is essential for promoting healthy child development. When families and schools work in concert to establish norms for SM use and consistently monitor adherence through regular communication, children’s life satisfaction and mental health outcomes can improve significantly [46]. Ongoing communication between parents and teachers is not only effective in preventing the negative effects of SM overuse but also facilitates a joint effort to address more complex psychological and behavioral issues, such as cyberbullying. Through this cross-sector collaboration, parents and educators can provide more comprehensive support for children, helping them maintain their mental health and social balance in an increasingly complex digital landscape [53].

### 4.5. Implications for Mental Health Professionals and Practitioners

The findings of this study emphasize the need for mental health professionals to adopt a multidimensional approach when assessing and intervening in children’s SM use. While high-intensity use (more than 2 h/day) is closely associated with psychological distress, suicidal ideation, and unmet mental health needs [40,58], greater attention should be paid to the patterns and motivations behind usage. Research shows that passive browsing of others’ content often leads to negative psychological effects, while active engagement can yield positive outcomes [40]. Therefore, practitioners need to distinguish between intensive use, problematic use, and specific behaviors (such as appearance comparison and seeking likes) in order to more accurately identify risks and develop interventions [48,50].

Additionally, different SM platforms have varying effects on mental health; for example, YouTube and Instagram users report more body image issues and eating pathology [50]. Hence, practitioners should familiarize themselves with the characteristics of each platform to identify high-risk individuals. Moreover, gender differences significantly influence the impact of SM use, with girls being more likely to experience negative effects due to appearance comparisons [50]. These gender differences suggest that practitioners should incorporate individualized considerations when developing interventions.

To adapt to the digital age, practitioners should continually enhance their digital health literacy and stay updated on the latest children’s usage patterns [41,48]. Improving children’s digital health literacy, particularly their ability to filter and assess online health information, can help them use SM more responsibly [41,46]. Furthermore, providing mental health support and resources through SM can reduce barriers to seeking help in real-life settings [43,58].

Finally, future research should focus on longitudinal analyses to clarify the causal relationship between SM use and CMH [42]. Further exploration of the specific impacts of different usage motivations and content on mental health, as well as the moderating roles of individual (e.g., self-esteem, self-compassion) and social factors (e.g., family support, peer relationships), will help to develop more targeted intervention strategies [41,54].

### 4.6. Short-Term Effects, Addiction Mechanisms, and Individual Differences

The short-term cognitive and emotional effects of SM on children warrant closer examination alongside its long-term consequences. The fragmented attention caused by constant notifications and updates disrupts focus, undermining academic performance and productivity. Additionally, the unceasing influx of information overloads cognitive processes, impairing memory encoding and recall, which negatively affects both educational and everyday tasks [46,50]. These short-term effects are intertwined with addiction mechanisms. SM platforms exploit the brain’s reward system by releasing dopamine in response to notifications or comments. This reinforcement fosters habitual usage and, in some cases, compulsive behaviors that hinder other developmental activities [50]. The cycle of distraction and reinforcement exacerbates these risks, promoting persistent checking behaviors that disrupt routines. The parental regulation of screen time has shown promise in mitigating these tendencies, emphasizing the importance of external interventions to curb excessive use [46,50].

Individual differences further shape how children experience SM. Gender disparities are pronounced, with girls disproportionately affected by cyberbullying and societal pressures related to appearance, resulting in lower self-esteem and higher rates of depressive symptoms [40,48]. Boys, in contrast, show greater resilience to these specific influences. Socioeconomic status also plays a pivotal role, as children from lower-income families often face limited access to digital resources, compounding their vulnerability to SM’s negative effects. Cultural factors further influence these dynamics, underscoring the need for tailored, context-sensitive interventions [45,54].

### 4.7. Limitations

This scoping review has several limitations and shortcomings. First, the studies employed diverse methodological designs (e.g., cross-sectional studies, longitudinal studies, technical reports, etc.), which may have led to a diminished comparability between the results and increased complexity in integrating and interpreting the findings. To address this issue, future reviews could benefit from a methodological quality assessment of the included studies, even if they vary widely in design. This additional step could help to mitigate potential biases and improve the reliability of the synthesized findings. Second, some studies suffered from insufficient sample sizes or sample selection bias, potentially affecting the accuracy and generalizability of the results. Additionally, the literature does not clearly indicate whether SM use directly contributes to the emergence of internalizing symptoms and problematic behaviors in children, or whether children with pre-existing psychological distress are more inclined to engage with SM. Although a correlation exists between the two, the causal relationship remains unclear.

Furthermore, despite the diligent efforts of this research to screen the relevant literature across multiple databases, some pertinent studies may have been overlooked due to limitations in database coverage and publication periods. Given the rapid evolution of SM technologies and changes in children’s usage behaviors, some early literature may not accurately reflect current trends, thereby affecting the timeliness and relevance of the findings.

## 5. Conclusions

This scoping review adhered to PRISMA-ScR guidelines and examined the impact of SM on CMH, specifically focusing on children aged 6 to 13 years. The review revealed that SM use has both positive and negative effects on CMH, with factors such as age, gender, social support, and parental involvement playing significant roles. The study emphasizes the importance of balancing SM use and highlights the necessity of robust family and school support systems to mitigate potential negative consequences. To promote healthy SM usage, practical parent training programs and school-based educational curricula are recommended to encourage collaboration between parents and educators.

## Figures and Tables

**Figure 1 healthcare-12-02391-f001:**
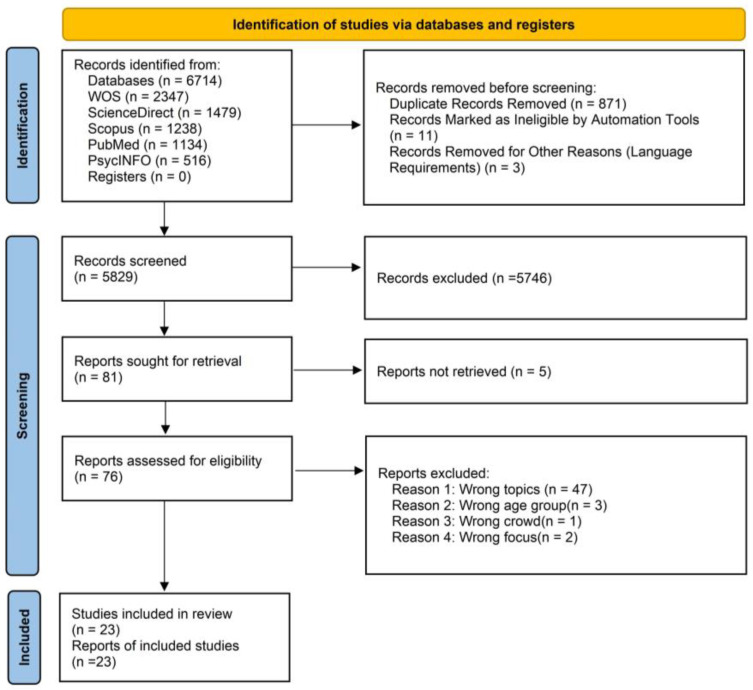
PRISMA flowchart.

**Figure 2 healthcare-12-02391-f002:**
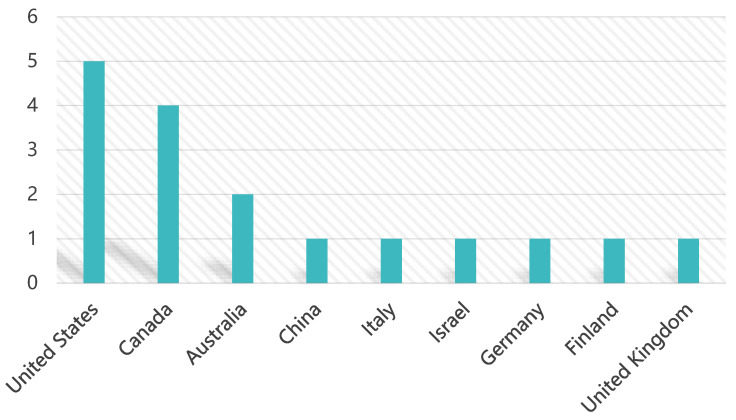
Countries of publications.

**Figure 3 healthcare-12-02391-f003:**
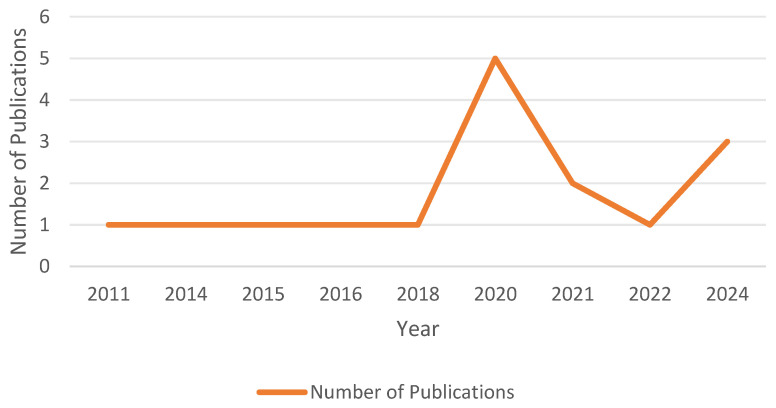
Year of publications.

**Figure 4 healthcare-12-02391-f004:**
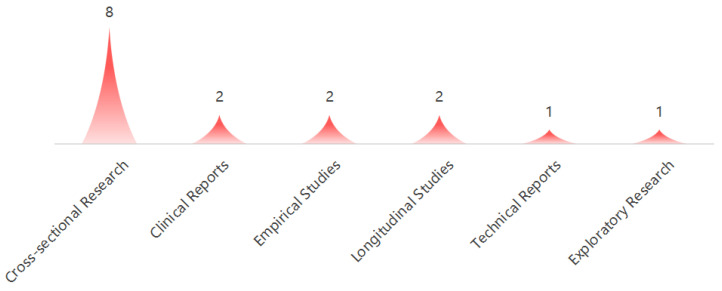
Types of studies.

**Figure 5 healthcare-12-02391-f005:**
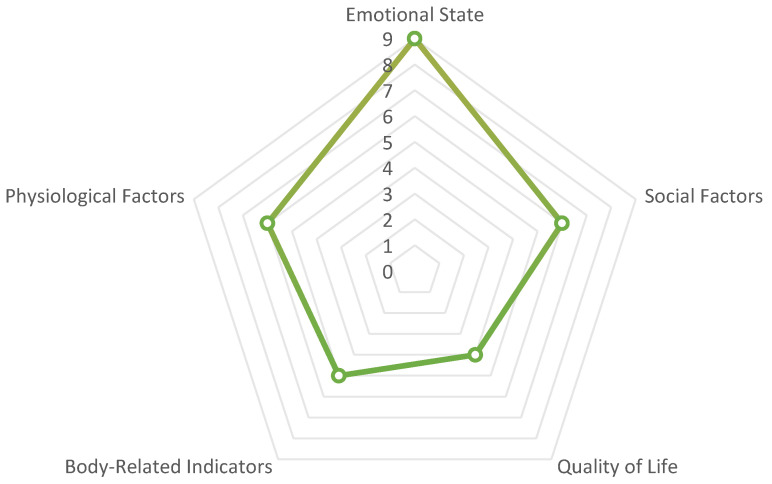
Visualization of related indicators.

**Figure 6 healthcare-12-02391-f006:**
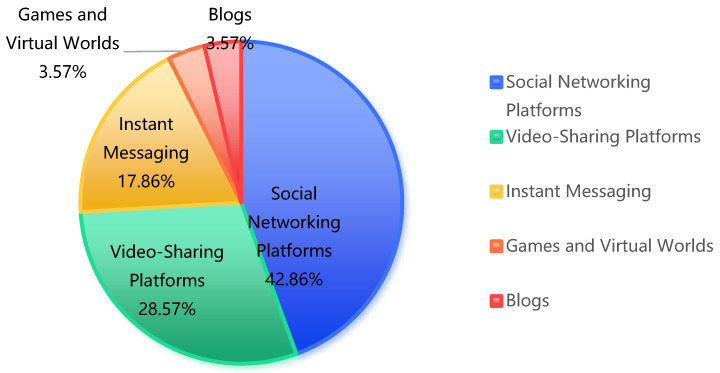
Percentage of SM platforms.

**Figure 7 healthcare-12-02391-f007:**
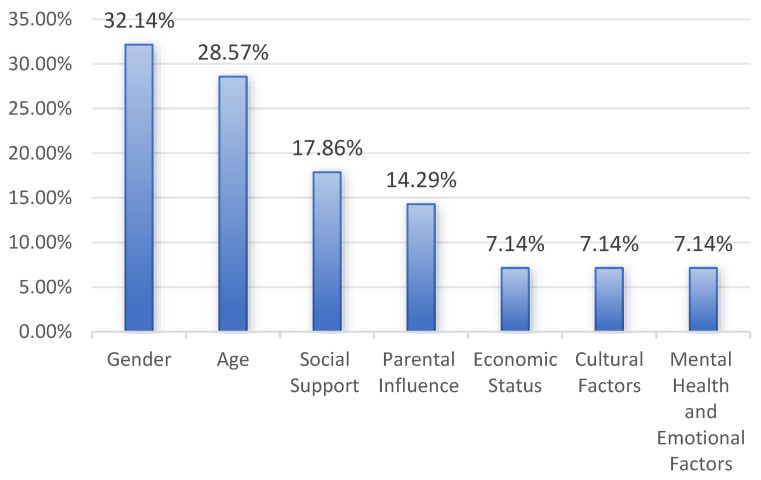
Percentage of factors/influences.

**Table 1 healthcare-12-02391-t001:** Selected databases and search formats.

Database	Search Formula
Web of Science	TS = (“social media” OR “new media” OR “online platform”) AND TS = (“children” OR “minors” OR “child”) AND TS = (“mental health” OR “well-being”) AND PY = (2014–2024)
ScienceDirect	(“social media” OR “new media” OR “online platform”) AND (“children” OR “minors” OR “child”) AND (“mental health” OR “well-being”) AND (2014 TO 2024)
Scopus	TITLE-ABS-KEY (“social media” OR “new media” OR “online platform”) AND TITLE-ABS-KEY (“children” OR “minors” OR “child”) AND TITLE-ABS-KEY (“mental health” OR “well-being”) AND PUBYEAR > 2013 AND PUBYEAR < 2025
PubMed	(“social media” OR “new media” OR “online platform”) AND (“children” OR “minors” OR “child”) AND (“mental health” OR “well-being”) AND (2014–2024)
PsycINFO	((“social media” or “new media” or “online platform”) and (“children” or “minors” or “child”) and (“mental health” or “well-being”)).mp. [mp = title, abstract, heading word, table of contents, key concepts, original title, tests & measures, mesh word] limit 2 to yr = “2014–2024”

**Table 2 healthcare-12-02391-t002:** Inclusion and exclusion criteria.

PICO Elements	Inclusion Criteria	Exclusion Criteria
Population (P)	Children aged 6–13 years, healthy individuals	Children with medical conditions or outside age range (>13 or <6)
Intervention (I)	Impact of social media on mental health	Studies not focusing on social media or related topics
Comparison (C)	Not applicable (no specific control group)	N/A
Outcome (O)	Psychological well-being, emotional impact, behavioral changes	Studies without a focus on mental health outcomes
Type of Study	Empirical research	Literature reviews, book chapters, theses, etc.
Publication Date	Published between 2014 and 2024	Published outside the 2014–2024 range
Language	Full text in English	Full text in other languages

**Table 3 healthcare-12-02391-t003:** Overview of study characteristics.

Author/Year/Country	Type	Topic	Indicator	Sample Size	Age	Area	SM Type	Factor	Main Findings
Fardouly [40] 2020 Australia	Cross-sectional Research	Depression	Body Image, Pathological Eating, Depressive Symptoms, Social Anxiety	528	10–12 years	Australia	Video-sharing platforms (e.g., YouTube), social networking platforms (e.g., Facebook, Instagram), instant messaging (e.g., Snapchat)	Gender, Appearance Comparison, Appearance Investment	SM use (especially YouTube, Instagram, and Snapchat) is linked to low body satisfaction and high eating pathology, with frequent appearance comparisons predicting declines in mental health.
Donelle et al. [41] 2021 Canada	Cross-sectional Research	Digital Literacy	Emotional State, Social Interaction	42	6–10 years	Canada	Video-sharing platforms (e.g., YouTube), social networks (e.g., Facebook, Instagram), instant messaging (e.g., Snapchat)	Age, Gender, Type of SM Platform Used, Device Type	57% of children use SM, with YouTube being the most popular; half share personal content but lack privacy awareness, posing potential privacy risks.
Marengo et al. [42] 2021 Italy	Cross-sectional Research	Cyberbullying	Self-esteem, Depression, Anxiety, Social Support	3022	11, 13, 15 years	Italy	Not specified	Gender, Age, Social Support (Family, School, Peers)	Girls have a higher proportion of online bullying victimization and problematic SM use, which are positively correlated, and social support can mitigate this association.
O’Keeffe et al. [43] 2011 United States	Clinical Reports	Social and Emotional Development	Depression, Anxiety, Severe Isolation, Suicide	Not provided	Various age groups	Not provided	Social networking platforms (e.g., Facebook, MySpace, Twitter), games and virtual worlds (e.g., *Club Penguin, Second Life, The Sims*), video platforms (e.g., YouTube), blogs	Parents’ Understanding of SM, Technical Ability, Communication with Children	SM offers opportunities for social interaction and emotional expression but also presents issues such as online bullying and privacy violations, requiring parental guidance and medical education.
Wong et al. [44] 2022 Canada	Cross-sectional Research	Interpersonal Relationships and Social Skills	Relationship Support, Isolation, Self-esteem	17,149	11–15 years	Canada	Video-sharing platforms (e.g., YouTube), social networks (e.g., Facebook), instant messaging (e.g., Snapchat)	Age, Gender, Household Economic Status	Healthy SM use strengthens friendships, while problematic use leads to poor family relationships and social isolation.
Chassiakos et al. [37] 2016 United States	Technical Reports	Depression	Sleep, Attention, Learning, Obesity, Depression	Not provided	6–18 years	Not provided	Video-sharing platforms (e.g., YouTube), social networks (e.g., Facebook), multiplayer video games, video blogs (Vlogs), etc.	Age, Gender, Social Support, Type of SM Use, Usage Duration, Cyberbullying, Family Media Use Behavior	Digital media have a dual impact on the mental health of children and adolescents; moderate use is beneficial, while excessive use is harmful, necessitating the establishment of healthy usage plans.
Guo et al. [45] 2024 China	Empirical Studies	Subjective Well-being	Depressive Symptoms, Self-esteem Level, Self-compassion	386	9–12 years	China	SM platforms (e.g., Twitter, Facebook, Instagram), social networking sites (e.g., WeChat Moments, Qzone, Sina Microblogs)	Self-Esteem, Self-Compassion, SNS Use Intensity and Experience	SNS use is positively correlated with depressive symptoms among children, with self-esteem and self-compassion playing regulatory roles.
Shoshani et al. [46] 2024 Israel	Longitudinal Studies	SM Risks	Depression, Anxiety, Psychological Distress, Life Satisfaction, Emotional State	3697	8–14 years	Israel	Video-sharing platforms (e.g., YouTube), social networking platforms (e.g., Facebook, Instagram), short video platforms (e.g., TikTok), instant messaging (e.g., WhatsApp, Snapchat)	Social Support, Extracurricular Activities, Age, Gender	Increased SM use leads to rising mental symptoms and declining well-being among children and adolescents, which can be alleviated by social support and extracurricular activities.
Richter et al. [47] 2020 Germany	Cross-sectional Research	SM Threats	Health Self-assessment, Psychosomatic Symptoms, Life Satisfaction, Risk Behavior	5094	11, 13, 15 years	Germany	SM platforms (e.g., Twitter, Facebook, Instagram), instant messaging (WhatsApp, Telegram, Snapchat)	Gender, Age, School Type, Immigrant Background	Frequent SM use is associated with poor self-health assessments among girls and reduced school satisfaction among boys, and is significantly related to smoking, drinking, and bullying behavior regardless of gender.
Lahti et al. [48] 2024 Finland	Cross-sectional Research	SM Anticipatory Guidance	Self-rated Health, Depressive Mood, Anxiety Symptoms	2288	11, 13, 15 years	Finland	Not specified	Gender, Age, Emotional Intelligence, Family Support, Friend Support	Children and adolescents are often exposed to misinformation and appearance pressure on SM, and problematic use increases threat exposure; frequent exposure is linked to poor self-rated health, depression, and anxiety, while high emotional intelligence and family support can reduce threat exposure frequency.
Hill et al. [49] 2020 United States	Clinical Reports	Parental Control	Anxiety, Depression, Self-esteem, Sleep, Weight Management	Not provided	0–18 years	Not provided	Social networking platforms (e.g., Facebook, Instagram)	Age Group Differences, Parental Monitoring Frequency, Screen Time Management, Online Social Behavior	SM have multiple impacts on children and adolescents; parental monitoring and guidance can reduce risks, and discussing SM use with doctors can improve health.
Fardouly et al. [50] 2018 Australia	Empirical Studies	Gender Equality	Depressive Symptoms, Appearance Satisfaction, Life Satisfaction	284	10–12 years	Australia	Social networking platforms (e.g., Facebook, Instagram)	Parental Control Methods, Mental Health Status	The less control parents have over their children’s SM usage time, the higher the frequency of appearance comparisons among children, which is associated with poorer mental health.
Nagata et al. [51] 2020 United States	Cross-sectional Research	Excessive Social Networking	Life Satisfaction, Mental Health	Not provided	11, 13, 15 years	45 countries	Not specified	Socioeconomic Factors, Gender Equality	Low social support and problematic SM use predict low life satisfaction; mental health among children in high-income countries declines, with girls facing increased school pressure, and insufficient sleep and problematic SM use are associated with poorer well-being.
Sampasa-Kanyinga et al. [52] 2015 Canada	Cross-sectional Research	Subjective Well-being	Self-rated Mental Health, Psychological Distress, Suicidal Ideation	753	11–18 years	Ottawa and Canada	Social networking platforms (e.g., Facebook, Twitter, MySpace, Instagram)	Daily Social Network Usage Time, Gender, Grade, Subjective Socioeconomic Status, Parental Education Level	Students who use SNS for more than 2 h per day have a significantly increased risk of mental health issues, including poor self-esteem, psychological distress, and suicidal ideation.
Twigg et al. [53] 2020 United Kingdom	Longitudinal Studies	Children’s Spirituality Development	Life Satisfaction	7596	10–15 years	United Kingdom	Social networking platforms (e.g., Facebook, Bebo, MySpace)	Gender, Family Support, Income	Frequent SM use is associated with changes in children’s life satisfaction, but is not the primary cause of deterioration; girls experience a more significant decline in well-being, and parental mental health, family support, and income significantly affect children’s life satisfaction.
Yust et al. [54] 2014 United States	Exploratory Research	Mental Health and SM Use	Spiritual Well-being, Relationship Experience, Emotional Development	Over 25,000	Not provided	Multiple European countries	Social networking platforms (e.g., Facebook, Twitter), online games	Cultural Background, Depth and Frequency of Digital Cultural Participation, Emotional Bond	Digital culture influences children’s identity and relational experiences; SM can both nurture children’s mental well-being and cause emotional disorders. While active participation can enhance extroversion and empathy, it may also lead to detached attachment.

**Table 4 healthcare-12-02391-t004:** Geographical distribution and sample characteristics.

Region	Number of Studies	Countries/Regions in Sample	Average Age	Sample Size Range
Europe	6	Italy, Germany, Finland, the UK, and Multiple European Countries	13 years	2288–over 25,000
North America	4	Canada	13 years	42–17,149
Asia–Pacific	3	Australia, China	10 years	284–528

**Table 5 healthcare-12-02391-t005:** Research topics.

Research Topics	Subtopics	Amount
Mental Health Issues	Depression	3
	Neurosis	
	Subjective Well-being	
Digital Literacy and Risk Awareness	Digital Literacy	3
	SM Risks	
	SM Threats	
Cyberbullying and Guidance	Cyberbullying	2
	SM Anticipatory Guidance	
Social and Emotional Development	Social and Emotional Development	2
	Interpersonal Relationships and	
	Social Skills	
Family and Society	Parental Control	2
	Gender Equality	

**Table 6 healthcare-12-02391-t006:** Indicators of the impact of SM on CMH.

Category	Indicator Items	Occurrence Counts
Emotional State	Depressive Symptoms, Social Anxiety, Emotional State, Anxiety, Depression, Self-esteem Level, Self-compassion, Psychological Distress, Appearance Satisfaction	9
Social Factors	Social Interaction, Social Support, Relationship Support, Isolation, Self-esteem, Severe Isolation	6
Physiological Factors	Attention, Learning, Suicide, Spiritual Well-being, Relationship Experience, Emotional Development	6
Body-Related Indicators	Body Image, Pathological Eating, Obesity, Sleep, Weight Management	5
Quality of Life	Life Satisfaction, Health Self-assessment, Risk Behavior, Mental Health	4

**Table 7 healthcare-12-02391-t007:** SM platforms found in the literature.

Platform Type	Examples	Frequency	Percentage
Social Networking Platforms	Facebook, Instagram, Twitter, MySpace	12	42.86%
Video-Sharing Platforms	YouTube	8	28.57%
Instant Messaging	Snapchat, WhatsApp, Telegram	5	17.86%
Games and Virtual Worlds	*Club Penguin, Second Life, The Sims*	1	3.57%
Blogs	N/A	1	3.57%

**Table 8 healthcare-12-02391-t008:** Factors and influences found in the literature.

N	Factors/Influences	%
9	Gender	32.14%
8	Age	28.57%
5	Social Support (family, school, friends)	17.86%
4	Parental Influence (understanding of SM, monitoring frequency, and control methods)	14.29%
2	Economic Status (household and subjective economic status)	7.14%
2	Cultural Factors (cultural background and depth of digital cultural participation)	7.14%
2	Mental Health and Emotional Factors (emotional intelligence and mental health status)	7.14%

## Data Availability

The authors confirm that all data generated or analyzed during this study are included in this published article. Furthermore, primary and secondary sources and data supporting the findings of this study were all publicly available at the time of submission.

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
