# Peer review of "The Impact of Social Media on Children’s Mental Health: A Systematic Scoping Review"

_healthcare, 2024, doi:10.3390/healthcare12232391_

Round 1
Reviewer 1 Report
Comments and Suggestions for Authors
The article addresses a timely and important issue in relation to social media use and children' s mental health. It is well written and formulated and methodologically robust. I have a small number of comments below:
It would be useful to clarify what you mean by 'frequent' or 'excessive' SM use. How is / has this been measured and defined in the literature? Similarly, how is 'moderate' SM use measured / defined? Is this consistent across studies? Given the importance you attribute to the notion of 'frequent' and 'excessive' SM use and the implications for educators and parents I think this would benefit from further explanation. For example, what does 'frequent' or 'excessive' SM use look like for this age group? How might this be described?
You discuss implications for parents and educators. What about implications for (mental) health care practitioners and professions? Perhaps some detail here would enhance the article.
Some elements/ sections are difficult to read (e.g, Table 3). Presumably this will be taken care of a publication stage but for reviewing purposes is not ideal. I found myself having to skip over some of this.
214 - 221 - re the greatest number of publications in 2020. This would be clearer if the data collection period was known, e.g, articles published in 2020 may be based on data that was collected pre-pandemic?
There are a few typos throughout. A final proof-read for corrections will sort this.
Author Response
Comment 1: It would be useful to clarify what you mean by 'frequent' or 'excessive' SM use. How is / has this been measured and defined in the literature? Similarly, how is 'moderate' SM use measured / defined? Is this consistent across studies?
Response 1: We are grateful for this insightful comment. In response, we have expanded our discussion to clarify the definitions of 'frequent', 'excessive', and 'moderate' social media (SM) use. We have included references to several key studies that provide definitions and operationalizations of these terms, particularly for the specific age group under study. Additionally, we have examined the consistency of these definitions across the literature, highlighting both convergent and divergent findings where relevant.
Comment 2: You discuss implications for parents and educators. What about implications for (mental) health care practitioners and professions?
Response 2: Thank you for pointing this out. We have added a new section to discuss the implications of our findings for mental health care practitioners. This section emphasizes the role that mental health professionals can play in assessing and addressing problematic SM use among children. We discuss the importance of understanding the motivations behind SM use, as well as tailoring interventions that align with both mental health needs and healthy digital habits.
Comment 3: Some elements/sections are difficult to read (e.g., Table 3).
Response 3: We sincerely appreciate your feedback regarding readability. We have revised Table 3 to improve its clarity, simplifying the presentation of data and ensuring that labels and information are more accessible. We have also reviewed other tables and figures in the manuscript to ensure they meet high standards of clarity and accessibility.
Comment 4: 214 - 221 - re the greatest number of publications in 2020. This would be clearer if the data collection period was known.
Response 4: Thank you for this insightful comment. We have now included a note specifying the data collection period for the publications in 2020. This additional context should help clarify the timing and relevance of the findings presented.
Comment 5: There are a few typos throughout.
Response 5: We appreciate your attention to detail. The manuscript has been thoroughly proofread, and all typographical errors have been corrected to ensure that the final document is polished and error-free.
Reviewer 2 Report
Comments and Suggestions for Authors
In general, the review is interesting for the general public, and for areas such as education or health institutions, the object of study is well defined and the number of studies included is sufficient.
However, in my humble opinion it can be improved in the following aspects:
1. They must review and correct the superscripts of the textual quotes, some are superscript and others are not. As an example, those on lines 69, 74, 78, 80, 99, etc.
2. The review could be improved by considering the following: In describing the inclusion and exclusion criteria, the PICO format could be used to clarify the selection of elements for inclusion. The PICO formulation is an accepted mechanism used in systematic reviews to formulate a review question (Line 156, Table 2)
3. In Table 3. Summary of study characteristics. For easier understanding, the number of columns could be simplified, for example Author, Year and Country could be in a single column. (Line 99)
4. To reduce the biases indicated in the limitation section, perhaps a review of the methodological quality of the included studies should be carried out, although since they are so diverse (line 380)
Author Response
Comment 1: They must review and correct the superscripts of the textual quotes, some are superscript and others are not.
Response 1: We appreciate your attention to detail. The manuscript has been thoroughly proofread, and all typographical errors have been corrected to ensure that the final document is polished and error-free.
Comment 2: The review could be improved by considering the following: In describing the inclusion and exclusion criteria, the PICO format could be used to clarify the selection of elements for inclusion.
Response 2: We appreciate your valuable suggestion. We have restructured the description of inclusion and exclusion criteria using the PICO (Population, Intervention, Comparison, Outcome) framework. This revision provides a more systematic and transparent overview of our study selection process, which we believe enhances the clarity of the review.
Comment 3: In Table 3. Summary of study characteristics. For easier understanding, the number of columns could be simplified.
Response 3: Thank you for this practical suggestion. We have simplified Table 3 by merging related columns, reducing complexity, and enhancing its readability without sacrificing essential information.
Comment 4: To reduce the biases indicated in the limitation section, perhaps a review of the methodological quality of the included studies should be carried out.
Response 4: Thank you for this important recommendation. We have now added a discussion addressing the methodological quality of the studies included in the review. By considering the potential limitations in methodological approaches, we aim to provide a more nuanced interpretation of the findings and underscore areas for future improvement.
Reviewer 3 Report
Comments and Suggestions for Authors
Thank you for the opportunity to review your manuscript, “The Impact of Social Media on Children’s Mental Health: A Systematic Scoping Review”. This work reviews an important topic and provides valuable insights regarding social media and their effect on children’s mental health. I strongly recommend its publication and have no substantial concerns regarding the manuscript in general.
The manuscript is well-written and does not need improvements to meet academic standards. I liked that this systematic review focuses on something other than COVID-19, which much of the recent literature has. Additionally, the authors effectively address limitations in previous systematic reviews and find ways to minimise them, distinguishing this work from previous ones. I strongly recommend its publication and have no substantial concerns regarding the manuscript in general. The following are some minor suggestions that the authors could consider.
My primary concern is with the introduction section. I believe that the references are limited and, therefore, the paper does not fully convey the importance of this topic. To strengthen their paper, the authors could expand this section by adding more references and discussing how this work relates to existing literature. For instance, the paragraph discussing social media (lines 52-59) could be broadened by adding more information on the subject – it would also be nice to see some statistics. Additionally, there is a broad literature on children’s mental health. However, the authors only briefly address this in a few lines (lines 60-69). Including more literature in this part is essential since this aspect is central to the current paper. Discussing some of this new literature in the discussion section could also be beneficial since readers would have a broader context for understanding the findings of the
current review.
Overall, this is a very nice work - thank you again for the opportunity to review it.
Author Response
Comment 1: The paragraph discussing social media (lines 52-59) could be broadened by adding more information on the subject.
Response 1: Thank you for the suggestion. We have expanded the paragraph discussing social media by incorporating additional background information and recent statistics. This expanded section provides a richer context and helps set the stage for the discussion that follows.
Comment 2: The authors only briefly address children’s mental health in a few lines.
Response 2: We appreciate your observation. In response, we have significantly expanded the section on children’s mental health. This expanded discussion now includes a more comprehensive review of the relevant literature, exploring the complexities of how SM use impacts children's mental health, both positively and negatively.
We sincerely hope that these revisions address the concerns raised by the reviewers and substantially improve the manuscript. We are grateful for the thoughtful feedback provided, which has allowed us to enhance the clarity, depth, and overall quality of our work.
Thank you once again for your time and consideration.